# Fast and Efficient Locomotion via Learned Gait Transitions

**Yuxiang Yang[1], Tingnan Zhang[2], Erwin Coumans[2], Jie Tan[2], and Byron Boots[1]**
[1]University of Washington, [2]Robotics at Google
{yuxiangy, bboots}@cs.washington.edu
{tingnan, erwincoumans, jietan}@google.com

**Abstract:** We focus on the problem of developing energy efficient controllers for quadrupedal robots. Animals can actively switch gaits at different speeds to lower their energy consumption. In this paper, we devise a hierarchical learning framework, in which distinctive locomotion gaits and natural gait transitions emerge automatically with a simple reward of energy minimization. We use evolutionary strategies (ES) to train a high-level gait policy that specifies gait patterns of each foot, while the low-level convex MPC controller optimizes the motor commands so that the robot can walk at a desired velocity using that gait pattern. We test our learning framework on a quadruped robot and demonstrate automatic gait transitions, from walking to trotting and to fly-trotting, as the robot increases its speed. We show that the learned hierarchical controller consumes much less energy across a wide range of locomotion speed than baseline controllers.

**Keywords:** Legged Locomotion, Hierarchical Control, Reinforcement Learning

## 1 Introduction

Fast and energy efficient locomotion is crucial for legged robots to accomplish tasks that traverse long distances. In the natural world, quadrupedal animals demonstrate a wide variety of distinctive locomotion patterns known as gaits [1], such as walking, trotting, bounding and galloping. Each gait is characterized by a unique foot contact schedule in a locomotion cycle. To lower their energy consumption, most quadrupedal animals switch to a preferred gait at each different speed range [1, 2]. While many locomotion gaits have been implemented on quadrupedal robots [3, 4], gait timing is often hand-engineered, and the switch between gaits is based on *ad-hoc* user commands. Can quadruped robots learn energy efficient gaits and natural gait transitions automatically?

In this work, we devise a learning framework in which energy-efficient locomotion controllers emerge automatically. The learned controllers naturally switch between different gaits at different speeds to maximize energy efficiency. Learning speed-adaptive locomotion gait controllers is challenging. Although reinforcement learning (RL) has been used to train policies end-to-end for a wide variety of continuous control tasks [5, 6, 7], these policies are often difficult to deploy safely on real robots without additional sim-to-real effort such as reward shaping [8], domain randomization [9, 10], or meta learning [11, 12]. Alternatively, optimal control based controllers have demonstrated robust performance on a number of quadruped robots [3, 13]. However, since gait patterns involve discrete contact events, it is difficult to optimize them together with other continuous forces. Therefore, most optimal control based controllers assume fixed, pre-selected gait timings.

To address the above challenges, we devise a hierarchical framework that combines the advantages of both RL and optimal control. This hierarchical framework consists of a high-level *gait generator* and a low-level *convex MPC controller*, which decouples the locomotion task into gait generation and motor control. Instead of directly outputting motor commands, the *gait policy* functions as part of the *gait generator* and specifies key gait parameters, which determines the contact schedule for each leg. Based on this contact schedule, the *convex MPC controller* then determines which legs are in contact, and computes the optimal motor command for each leg. We formulate the high-level gait policy learning as a Markov Decision Process (MDP), design a simple reward function based on velocity tracking and energy efficiency, and train the gait policy using evolutionary strategies (ES).

5th Conference on Robot Learning (CoRL 2021), London, UK.

We formulate the low-level convex MPC controller using model-predictive control with simplified dynamics [3].

With this hierarchical framework and the simple reward function, the gait policy automatically learns distinctive gait patterns at different locomotion speeds, including slow walking, mid-speed trotting and fast fly-trotting. Moreover, the policy automatically transitions from one gait to another to generate the most efficient gait at all speeds. Thanks to the robustness of our hierarchical framework, the learned gait policy can be deployed successfully on a Unitree A1 robot [14] in various environments (e.g. carpet, grass, short obstacle) without additional data collection and fine-tuning.

The main contributions of this paper include:

- A hierarchical learning framework effectively combines RL with optimal control, which can automatically learn fast and efficient locomotion controllers;
- The learned controllers switch gaits across a wide range of locomotion speeds, similar to those demonstrated in the animal kingdom;
- The learned controller can be deployed directly to the real world, and performs robustly in various environments.

## 2  Related Work

Quadrupedal animals demonstrate a wide variety of gaits [15]. Hoyt and Taylor [1] showed empirically that horses minimize their energy consumption when using the preferred gait at each speed. Alexander and Jayes [2] generalized this result by providing a unifying theory of gait transitions for quadrupedal animals. A wide variety of these gaits have been implemented in quadrupedal robots, including walking [16], trotting [3], pacing [4], bounding [17] and galloping [18]. In these works, model-based controllers [3, 19] optimize for motor commands at a high frequency. These controllers usually assume a pre-defined contact sequence to keep the optimization problem tractable, which does not allow gait transitions. Alternatively, contact implicit optimization [20, 21, 22] optimizes contact forces and sequences together, but is not feasible for real-time use due to high computation cost. Using manually designed heuristics, Boussema et al. [23] achieved online gait transition by computing the Feasible Impulse Set for each leg, at a speed up to 0.6m/s, or 1 body length/s. Owaki and Ishiguro [24] also demonstrated online gait transition on a 2kg quadruped robot using foot-force heuristics. Compared to these approaches, our learning-based approach requires less manual tuning, achieves more agile motions (up to 2.5m/s, or 5 body length/s) on a larger robot (15kg).

Recently, reinforcement learning became a popular approach to learn locomotion policies for legged robots [9, 25, 26]. Since policies are often learned in simulation, extra effort is usually required to transfer the learned policies to the real robot, including building more accurate simulation [9, 25], dynamic randomization [9, 10], motion imitation [27, 28] and meta learning [11, 12]. Inspired by the periodicity of locomotion behaviors, several methods have been proposed to make the learned policy more predictable and safer for real robot deployment, such as cyclic trajectory generators [29], phase-functioned neural networks [30] and state machines [31]. In contrast to these previous works, our learned policy achieves zero-shot sim-to-real transfer, and the controller is robust in multiple real-world environments.

Compared to directly learning an end-to-end controller, hierarchical learning [32] can improve data efficiency, and achieve complex tasks. The low-level controller can be a learned policy [33, 34, 10], or a hand-tuned controller [35, 36, 37]. Li et al. [36] uses a learned policy to modulate objectives of the low-level MPC controller, with a fixed contact sequence. Recently, Da et al. [35] looked into learning hierarchical controllers for gait control in quadrupeds, where the low-level controller is model-based and high-level policy selects from a fixed set of gait primitives. We use a similar hierarchical setup as Da et al. [35]'s, but extend the high-level policy to search a continuous range of gaits with arbitrary gait changes, and achieve significantly faster walking.

## 3  A Hierarchical Framework for Gait Optimization

### 3.1  Overview

To learn fast and efficient locomotion, we build a hierarchical framework with a high-level gait generator and a low-level convex MPC controller (Fig. 1). The high-level gait generator includes a

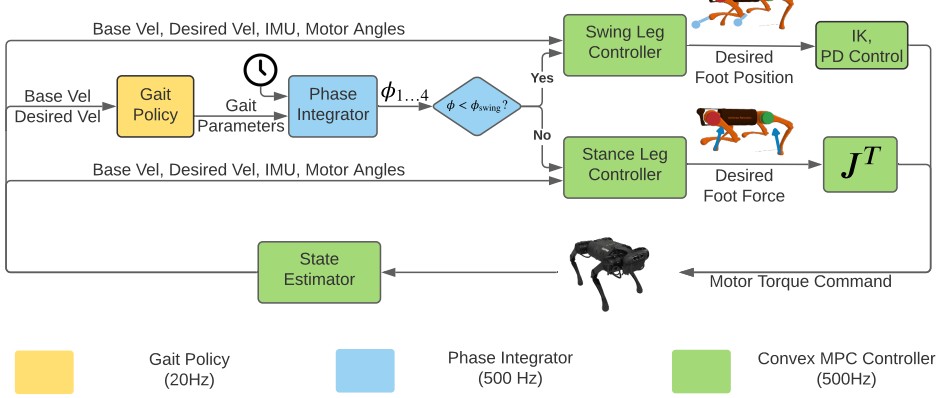

Figure 1: Our system consists of a high-level gait generator and a low-level convex MPC controller.

learnable gait policy and a phase integrator. To generate a gait, the gait policy outputs gait parameters, such as frequency, to the *phase integrator*. Based on these parameters and the robot's clock, the phase integrator increments the phase for each leg and determines its contact state. In each locomotion cycle, the phase progresses from $0$ to $2\pi$ as the foot goes from liftoff to touchdown to the next liftoff. The low-level convex MPC controller consists of separate controllers for swing and stance legs, and controls each leg differently based on its contact state. Additionally, we implement a Kalman Filter-based state estimator for the torso velocity, which cannot be measured directly using onboard sensors and is used by both the gait generator and the convex MPC controller. We run the high-level gait generator at 20Hz to avoid abrupt changes of gait commands, and the low level controllers at 500Hz for fast replanning and stable torque control.

## 3.2   High-Level Gait Generation

A locomotion gait is determined by a *contact schedule*, the time and duration that each leg is in contact with the ground. To generate foot contact schedules, the phase integrator maintains a set of phase variables $\phi_{1,\dots,4}$, one for each leg. The phase $\phi_i \in [0, 2\pi)$ denotes the leg's progress in its current gait cycle (Fig. 2). Each leg $i$ starts with *swing* at the beginning of a gait cycle ($\phi_i = 0$). As $\phi_i$ increases monotonically, it switches to *stance* after a threshold $\phi_i > \phi_{\text{swing}}$. After $\phi_i$ reaches $2\pi$, it wraps back to zero, and starts a new gait cycle in the *swing* phase. The propagation of phase variables, as well as the transition from *swing* to *stance*, are controlled by three key parameters, including stepping frequency $f$, swing ratio $p_{\text{swing}}$ and phase offsets $\theta_2, \theta_3, \theta_4$, which are specified by the gait policy. We choose such parameterization because it is expressive enough to represent a rich set of locomotion gaits. We now describe these parameters in detail:

**Stepping Frequency**   As a notable feature in locomotion, the stepping frequency is usually adjusted as a trade-off between speed and efficiency. While a high stepping frequency allows the robot

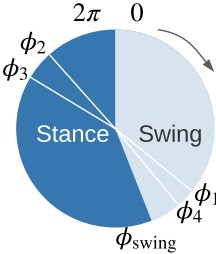

Figure 2: We use phases to represent the state of each leg. Each leg is assigned with an independent phase variable $\phi \in [0, 2\pi]$. $\phi_{\text{swing}}$ is the phase threshold that a leg switches from swing to stance. In the figure above, front-right ($\phi_1$) and rear-left ($\phi_4$) legs are in swing, while front-left ($\phi_2$) and rear-right ($\phi_3$) legs are in stance.

to run faster, stepping unnecessarily fast can cause additional energy consumption due to excessive leg swing. In our setup, the gait policy outputs the desired leg frequency $f \in (0\text{Hz}, 4\text{Hz}]$, which is used to increment the phase variable in the phase integrator. Specifically, at each control step, the phase is advanced by:

$$\phi[n] \leftarrow \phi[n-1] + 2\pi f \Delta t \tag{1}$$

where $\Delta t$ is the time step of low-level controller (0.002s).

**Swing Ratio**    Another important characteristic of gaits is the proportion of swing time in each gait cycle. For example, while a walking gait is usually characterized by spending less than 50% of a gait cycle in swing phase, a running gait typically requires a much longer swing time. To model this, we define another variable, $p_{\text{swing}} \in (0, 1)$, which controls the switching point in phase $\phi_{\text{swing}} = 2\pi p_{\text{swing}}$ between swing ($\phi < \phi_{\text{swing}}$) and stance ($\phi \geq \phi_{\text{swing}}$) in each gait cycle. Assuming that a leg moves at constant frequency, a larger $p_{\text{swing}}$ means that it will spend more time in air in a gait cycle, which usually results in a more dynamic gait.

**Phase Offsets**    Apart from careful design of individual gait cycles, careful coordination among legs is another critical component for efficient locomotion. While we use the same $f$ and $p_{\text{swing}}$ across all legs, we allow each individual leg to have a different phase offset. Let $\theta_i \in [0, 2\pi]$ denote the phase offset of leg $i$ compared to first leg (the front-right leg); then the phase of leg $i$ is $\phi_i = \phi_1 + \theta_i$. Note that the order of legs is [front-right, front-left, rear-right, rear-left], or [FR, FL, RR, RL] for short. For example, setting $\theta_4 = 0$ would make the rear-left leg in sync with the front-right leg, which is frequently seen in trotting gaits.

### 3.3 Low-Level Convex MPC Control

The low-level convex MPC controller computes and applies torques for each actuated degree of freedom, given the leg phases from the high-level gait generator. Our low-level convex MPC controller is based on Di Carlo et al. [3]. We briefly describe the controller here for the completeness of the paper. Please refer to the Appendix A for more details.

**Stance Leg Control**    In the stance leg controller, we model the robot dynamics based on the Centroidal Dynamics Model [3], where the full robot is simplified as a rigid-body base with massless legs. Each stance leg can generate ground reaction force at the contact point, subject to torque limit and friction cone constraints. These ground reaction forces are solved as a short-horizon MPC problem whose objective is for the robot base to closely track a given reference trajectory. We generate the reference trajectory based on user-specified velocity commands. The optimized contact forces $f$ are then converted to motor torques using the Jacobian transpose method: $\tau = \mathbf{J}^T f$. In our MPC setup, we re-optimize for ground reaction forces every time step (2ms), and only apply the first command in the optimized sequence.

**Swing Leg Control**    The swing leg controller calculates the swing foot trajectories and uses Proportional-Derivative (PD) controllers to track these trajectories. The swing trajectory is computed by fitting a quadratic polynomial over the lift-off, mid-air and landing position of each foot, where the lift-off position is the foot location at the beginning of the swing phase, the landing position is calculated using the Raibert Heuristics [38], and the mid-air location is set to ensure the minimum ground clearance. Please refer to Appendix A.2 for more details. Given the position in the swing trajectory, we convert it to the desired motor position using inverse kinematics, and apply motor torques using PD controllers.

## 4    Learning Gait Policies for Fast and Efficient Locomotion

Since locomotion gait involves discrete contact events, it is difficult to model and optimize them together with other continuous forces. Instead, we formulate a Markov Decision Process and apply Evolutionary Strategies (ES) to discover the most energy efficient gaits at different speeds.

### 4.1    Preliminaries

The reinforcement learning problem is represented as a Markov Decision Process (MDP), which includes the state space $\mathcal{S}$, action space $\mathcal{A}$, transition probability $p(s_{t+1}|s_t, a_t)$, reward function

$r : \mathcal{S} \times \mathcal{A} \mapsto \mathbb{R}$, and initial state distribution $p_0(s_0)$. We aim to learn a policy $\pi : \mathcal{S} \mapsto \mathcal{A}$ that maximizes the expected cumulative reward over an episode of length $T$, which is defined as:

$$J(\pi) = \mathbb{E}_{s_0 \sim p_0, s_{t+1} \sim p(s_t, \pi(s_t))} \sum_{t=0}^{T} r(s_t, a_t) \tag{2}$$

## 4.2 MDP Formulation

In our MDP formulation, we only learn the gait policy in the high-level gait generator, and consolidate the other components, including phase integrator, the convex MPC controller, and the robot dynamics, into the environment. At each step, the gait policy outputs gait parameters and receives a reward, which is based on energy consumption and speed-tracking performance.

**State and Action Space**  Since we aim to optimize the gaits based on the current and desired speed, we only include the desired and actual linear velocity of the base in the state space $\mathbf{s} = [\bar{v}_{\text{base}}, v_{\text{base}}]$. We do not include prioperceptive information, such as joint angles and IMU readings, because the detailed control of balance and locomotion, which consumes these information, is delegated to the low-level controller. The action space is a 5-dimensional vector $\mathbf{a} = [f, p_{\text{swing}}, \theta_2, \theta_3, \theta_4]$, as defined in Section 3.2.

**Reward Design**  We design the reward function as a linear combination of survival bonus, speed-tracking penalty, and energy penalty:

$$r = c - w_v \underbrace{\left\| \frac{\bar{v}_{\text{base}} - v_{\text{base}}}{\bar{v}_{\text{base}}} \right\|^2}_{\text{Speed Penalty}} - w_e \underbrace{\frac{\sum_{i=1}^{12} \max(\tau_i \omega_i + \alpha \tau_i^2, 0)}{mg\bar{v}_{\text{base}}}}_{\text{Energy Penalty (Cost of Transport)}} \tag{3}$$

The survival bonus $c$ prevents the learning algorithm from falling into the local minima of early termination. The speed penalty is the L2 norm of the relative error between the desired ($\bar{v}_{\text{base}}$) and actual speed ($v_{\text{base}}$) of the base. The energy penalty estimates the Cost of Transport (CoT), a standard metric for measuring the efficiency of locomotion [24, 39, 40]. The numerator estimates total power consumption in all 12 motors based on the angular velocity ($\omega_i$) and torque ($\tau_i$) of each motor, and the motor parameter ($\alpha = 0.3$). (See Appendix B for details.) The denominator consists of the mass of the robot ($m = 15$kg), and gravity constant ($g = 9.8$m/s$^2$). In each episode, the desired velocity $\bar{v}_{\text{base}}$ starts at 0 m/s, increases linearly to 2.5m/s at 1m/s$^2$ and stays at 2.5m/s for the rest of the episode. We use the same weights $c = 3, w_v = 1, w_e = 0.37$ for all of our experiments.

**Early Termination on Infeasible Gaits**  Despite the robustness of low-level whole body controller, the robot can still lose balance if the gait is infeasible, such as standing with one leg for an extended amount of time. To avoid unnecessary exploration in suboptimal gaits, we terminate an episode early if the robot falls (i.e. orientation deviates significantly from the upright pose, or the robot's height becomes too low).

## 4.3 Policy Representation and Training

We represent our policy as a neural network with one hidden layer of 256 units and tanh nonlinearities. We chose this network architecture because it is sufficiently expressive to learn different gaits, and structurally compact to be efficiently optimized by our learning algorithm. We train our policies using Covariance Matrix Adaptation Evolution Strategy (CMA-ES) [41], a simple, parallelizable evolutionary algorithm that has been successfully applied to locomotion tasks [42, 43, 44]. Compared to other RL algorithms, CMA-ES performs exploration in the policy parameter space and does not require accurate value function estimation at every step [45, 46], which is well-suited for our complex hierarchical system. See Appendix C.1 for details.

## 5 Results and Analysis

We design experiments to validate that our framework can learn fast and efficient locomotion controllers. Particularly, we aim to answer the following questions:

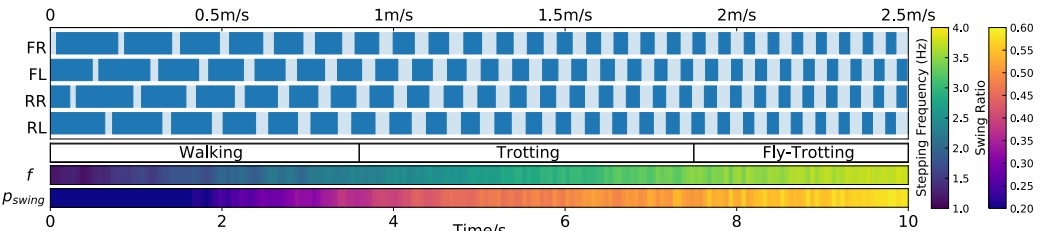

Figure 3: Robot gait during acceleration. Dark blue indicates foot contact. The robot switches from low-speed walking, to mid-speed trotting, to high-speed fly-trotting and increases the stepping frequency $f$ and swing ratio $p_{\text{swing}}$.

| | Freq (Hz) | Swing Ratio | Phase |
|---|---|---|---|
| Walk | 2 | 0.3 | $[0, \pi, 1.5\pi, 0.5\pi]$ |
| Slow Trot | 2 | 0.5 | $[0, \pi, \pi, 0]$ |
| Rapid Trot | 4 | 0.5 | $[0, \pi, \pi, 0]$ |
| Fly Trot | 4 | 0.6 | $[0, \pi, \pi, 0]$ |

Table 1: Parameters of hand-tuned gaits shown in Fig. 4. The order of leg phases is [FR, FL, RR, RL].

Figure 4: Cost of Transport (CoT) of hand-tuned gaits and our learned gait policy at different speeds. The CoT of learned gait is averaged over 5 random seeds. The standard deviation is too small to be visible in the figure.

1. Does our framework enable the robot to learn energy-efficient locomotion controllers?

2. Does the gait switching behavior, which is widely observed in animals, emerge naturally in the learning process?

3. Can the gait policy learned in simulation be deployed on real robots?

4. What are the advantages of our hierarchical framework and what are important design decisions?

## 5.1 Experiment Setup

We use the Unitree A1 robot [14], a small-scale, 15kg quadruped robot with 12 degrees of freedom. We use PyBullet [47] to simulate the robot dynamics. We implement the state estimation and high-level policy inference in Python, and the low-level Centroidal Dynamics-based Convex MPC Controller in C++. We use a Mac-Mini with M1 processor as our on-board computer, which runs the high-level gait generator at 20Hz and low-level convex MPC controller at 500Hz.

## 5.2 Emergence of Energy-Efficient Gaits

To demonstrate that our framework can learn efficient locomotion controllers, we evaluate the learned gait policy under different speed commands, and compare the Cost of Transport (CoT) with four carefully-designed, animal-inspired gaits: Walk, Slow Trot, Rapid Trot and Fly Trot. To find the parameters of these gaits, we first choose the phase offsets and swing-ratio based on the characteristics of each gait, and then perform grid search on stepping frequency to maximize the reward (Eq. 3) at different speed ranges. The parameters of these manually-designed gaits are summarized in Table. 1. We plot the CoT of each manually-designed gait and our learned controller (averaged over 5 random seeds) in Fig. 4. Clearly, the learned policy consistently achieves the lowest energy consumption at most of the speeds.

A closer look at Fig. 4 reveals that walking gait is the most efficient when the speed is below 0.8m/s, the slow and rapid trotting gait are the most efficient between 0.8 and 2m/s, and the fly-trotting gait is the most efficient when the speed is above 2m/s. We execute the learned gait policy on an accelerating speed profile, and are glad to find that the policy switches gaits at similar boundaries (Fig. 3). At low speeds (less than 0.9m/s), the policy exhibits a four-beat *walking* gait by moving one leg at a time in the order of [RR, FL, RL, FR]. At intermediate speeds (between 0.9 and 1.8 m/s), the policy synchronizes the diagonal legs and exhibits a *trotting* gait. At the highest speeds

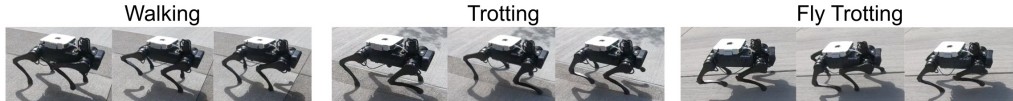

Walking        Trotting        Fly Trotting

Figure 5: Real world deployment of the learned policy. The robot moves at different speeds with different gaits.

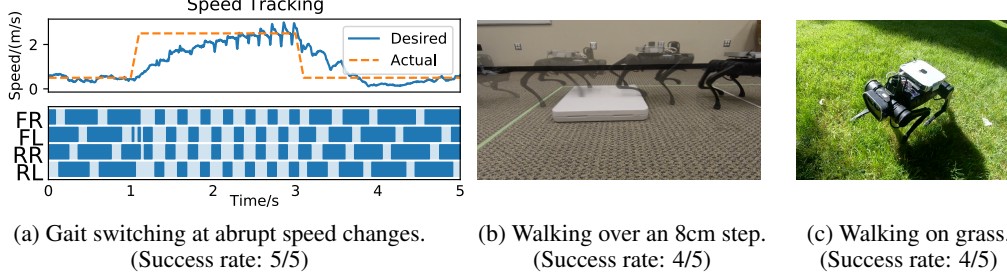

(a) Gait switching at abrupt speed changes. (Success rate: 5/5)      (b) Walking over an 8cm step. (Success rate: 4/5)      (c) Walking on grass. (Success rate: 4/5)

Figure 6: Deployment of gait policy to novel scenarios not encountered during training.

(1.8m/s or above), the policy exhibits a *fly-trotting* gait, with noticeable "airborne-phases" when all legs lift from the ground (characterized by $p_{\text{swing}} > 0.5$).

Compared to gaits observed in quadrupedal animals, our gait policy discovered similar behaviors at low and mid-range speeds (walking and trotting). However, at high speeds, many animals would *gallop*. To understand why our method does not automatically learn galloping at higher speeds, we reproduced a galloping gait in our framework by limiting the range of phase offsets. However, we found the learned galloping gait to be actually 30% *less* efficient than the flying trot on our robot, in contrast to animals [1]. This difference between animals and robots has been noted in [39], and it was hypothesized that such difference could result from differences in morphology, kinematics limits and joint actuation.

## 5.3 Validation on the Real Robot

We deploy our hierarchical controller, including the learned gait policy and the low-level model-based controllers, to the real robot (Fig. 5). Please watch the accompanying video. In contrast to many RL works that focus on sim-to-real transfer [9, 11, 12], our gait policy, learned entirely in simulation with PyBullet, can be directly deployed to the real world without additional data collection or fine-tuning. Similar to the simulation results, as the robot accelerates, it dynamically switches between walking, trotting and fly-trotting gaits, and eventually reached the speed of 2.5m/s, or 5 body lengths per second.

We test the generalization of our learned controller in a number of novel scenarios that were not seen during training. Although the policy is only trained using a slowly accelerating desired speed profile, the robot remains stable with abrupt changes of the desired speed, such as sharp acceleration and braking (Fig. 6a). The learned controller also generalizes to new terrains, including walking over an 8-cm step (Fig. 6b) and on grass (Fig. 6c). This excellent generalization is attribute to our hierarchical setup with a robust low-level convex MPC controller, which has demonstrated proven robustness on a wide variety of robot systems [3, 48, 49].

## 5.4 Comparisons with Non-Hierarchical Policies

To demonstrate the importance of the hierarchical setup, we compare the performance of our hierarchical framework with two non-hierarchical baselines: E2E and PMTG. E2E is a fully-connected end-to-end policy that directly maps from state to motor actions. PMTG implements the Policies Modulating Trajectory Generator [29], which simplifies learning by incorporating cyclic motion priors, and is widely used in the locomotion learning literature [33, 50, 51]. Following the prior work [52], we expand the state space to include IMU readings and motor angles, and modify the action space to be desired motor positions. Additionally, we carefully tuned the reward function in each baseline for optimal performance. See Appendix C.2 for details. We train all policies using the same CMA-ES algorithm.

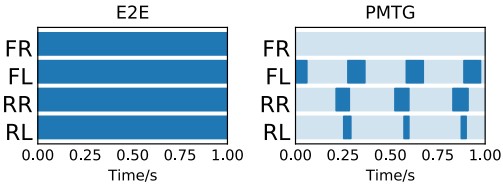 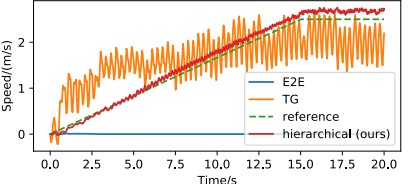

(a) Gaits Learned by Non-Hierarchical Policies.      (b) Speed Tracking Performance.

Figure 7: Gaits and speed-tracking performance of learned non-hierarchical policies that directly output motor commands. **Left**: The E2E policy falls into the local minima and stands in-place without moving. The PMTG policy moves forward using three legs. **Right**: Both policies do not track the desired speed as well as our hierarchical framework.

We find that E2E only learns to stand, while PMTG learns an unnatural gait of using 3 legs, and cannot track the desired speed well (Fig. 7). This is likely due to the optimization landscape of the flat policy parameterization being more jaggy and the optimization converges to one of the bad local optima. While additional reward shaping could yield better results, our hierarchical system provides a simple alternative to effectively learn locomotion policies.

## 5.5 Comparisons with Different Learning Algorithms

Using Evolutionary Strategies (ES) to train our hierarchical controller is a critical design decision. To understand its importance, we compare CMA-ES with other state-of-the-art methods, including PPO [53], SAC [7] and ARS [5] in training the hierarchical controller. Since both PPO and SAC can benefit from more comprehensive state information, we also run these algorithms with an extended observation space, which includes IMU readings, motor angles and gait phases, in addition to current and desired velocities. Please refer to Appendix C.1 for setup details and additional results.

As shown in Table. 2, algorithms based on Evolutionary Strategies (ES), CMA-ES and ARS, significantly outperforms other algorithms. When using the original observation space, both PPO and SAC fail to complete the task, which is likely due to the lack of sufficient information to accurately estimate the value function. With the extended observation space, SAC learns to walk forward, but fails to track the speed closely, and consumes more energy, compared to ES-based algorithms. We hypothesize that the poor performance of these two popular reinforcement learning algorithms is because the low-level convex MPC controller is a black box to the RL agent, which is not fully-observable and make the environment less Markovian.

| Algorithm | Success? | CoT | Avg Speed Tracking Error |
|---|---|---|---|
| PPO | No | $2.29 \pm 0.96$ | $0.080 \pm 0.092$ |
| PPO (extended obs) | No | $3.60 \pm 1.05$ | $0.15 \pm 0.17$ |
| SAC | No | $2.22 \pm 0.47$ | $0.069 \pm 0.025$ |
| SAC (extended obs) | Yes | $1.19 \pm 0.04$ | $0.030 \pm 0.025$ |
| ARS | Yes | $\mathbf{0.87 \pm 0.0073}$ | $\mathbf{0.0082 \pm 0.00034}$ |
| CMA-ES (ours) | Yes | $\mathbf{0.84 \pm 0.017}$ | $\mathbf{0.0083 \pm 0.00075}$ |

Table 2: Cost of Transport (CoT) and speed tracking error (Eq. 3) for gait policies trained by different algorithms. Error bar shows 1 standard deviation.

## 6 Conclusion

We present a hierarchical learning framework that can automatically learn fast and efficient gaits for quadrupedal robots. Our framework combines a high-level gait generator with a low-level convex MPC controller, where a gait policy is trained using evolutionary strategies with a simple reward function. Through learning, distinctive gaits and natural transitions between gaits emerge automatically. More importantly, the policy learned in simulation can be successfully deployed to the real world, thanks to the hierarchical setup. Observations in our robotic experiments agree well with prior bio-mechanical studies, which showed that quadrupedal animals switch gaits at different speeds to lower their energy expenditure. Our hierarchical control framework is general, and can be extended to modulate not only the gait patterns, but also other parts of the low-level controller, such as foot placement positions and desired base pose, to enable more agile and versatile locomotion skills. We plan to further develop this hierarchical framework through the lens of bi-level optimization, and apply it to other robotic platforms.

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
