# OpenReview forum: "Fast and Efficient Locomotion via Learned Gait Transitions"
_robot-learning.org/CoRL/2021/Conference — CoRL2021 Oral_

### Official Review · Reviewer_22ve · 2021-07-17

**Originality:** Fair
**Technical Quality:** Good
**Clarity Of Presentation:** Very Good
**Impact:** 3

**Recommendation:**

Strong Accept: I recommend accepting the paper and will argue for my recommendation even if other reviewers hold a different opinion.

**Summary:**

This paper presents a hierarchical framework for building locomotion controllers that adaptively change the walking gait at different speeds. I liked that the system design is straightforward, the experiments are interesting, and the results look good. The design of the high-level gait generation task is very similar to other related work such as (1) phase-functioned neural networks and (2) simbicon (which are missing from the references). Plus, the low-level controller is based on some previous work (Di Carlo et al. [3]). These made me realize that, although this is a nice work with good-looking results, it's incremental and requires more novelty to be published in CoRL. Plus, framing CMA-ES as a RL algorithm seems misleading to me. So for now, I prefer to vote for a weak reject.

**Issues:**

- In Eq. (3), it's not clear to me how the reward always stays positive. For example, the speed penalty can easily take a very large value if the robot starts from the idle state, where the actual speed (v_base) is almost zero. Later in Section 4.3 it's explained how the desired velocity is increased linearly after the episode starts, but (in my opinion) during the training the agent isn't able to follow the same speed pattern. So what happens if the actual speed is 0 (or 0.01) and the desired speed is 2.5?
- In Section 4.3, it's mentioned that the policy is trained using CMA-ES and the authors compare it to "other" RL algorithms. But CMA-ES is not a RL algorithm.
- In Section 4.3, the authors do not report the CMA-ES training configuration. It is a very important thing to miss, considering the fact that this is the only learning component of the system.
- For compuing the speed and energy penalties in Eq. (3), I think it would be better if the authors use exponential reward functions as, for example, explained in Section 5.3 of the DeepMimic paper:
Peng, X.B., Abbeel, P., Levine, S. and van de Panne, M., 2018. Deepmimic: Example-guided deep reinforcement learning of physics-based character skills. ACM Transactions on Graphics (TOG), 37(4), pp.1-14.
- It might not be a fair criticism since it's later explained by the authors, but in Section 5.2, I'd prefer to analyze Figure 3 differently. Animals optimize their energy consumption by changing their gait in different velocities. So, the correct plot in Figure 3 should have only one curve for the hand-tuned gaits, such that it uses the minimum CoT among the four hand-tuned gaits (or a linear combination of them). Then it would be observed that CoT of the learned gait and the hand-tuned one is almost identical. I believe this is what the paper was looking for in the first place (to show that their system can optimize the energy consumption in different speeds).
- It seems that in Figure 6.a, the labels for the desired and the actual velocities are misplaced. It does not make sense to me that the curve for actual speed is piecewise constant.
- In Section 5.5, the authors have done a poor job reporting the experiments using other algorithms. It is very important to report the number of timesteps, the size of the policy network, learning rate, etc. and also show the learning curves of the algorithms.
- Watching the video, it's difficult to see the difference between trotting and fly-trotting. It would have been better to also know about the failure cases (both in the video and in the paper).
- I expected to see the following papers in the references, since they use similar designs for modelling locomotion controllers:
Holden, D., Komura, T. and Saito, J., 2017. Phase-functioned neural networks for character control. ACM Transactions on Graphics (TOG), 36(4), pp.1-13.
Yin, K., Loken, K. and Van de Panne, M., 2007. Simbicon: Simple biped locomotion control. ACM Transactions on Graphics (TOG), 26(3), pp.105-es.

Typos:
- Line 15: traverses -> traverse
- Line 21: gates -> gaits
- Line 220: foot frequency -> stepping frequency?

**Reviewer Expertise:**

Very good: Comprehensive knowledge of the area

**Strengths And Weaknesses:**

Strengths:
- The design is straightforward. I like the formulation of the high-level gait generation task. The low-level controller also seems robust.
- The paper is well-written and it's easy to read and follow.
- Video results are looking good. However, it's difficult to see the difference between trotting and fly-trotting.
Weaknesses:
- I believe the paper is misleading the reader by claiming that CMAES is a RL algorithm. If we accept this claim, doesn't this make any other classical search or optimization algorithm a RL algorithm, assuming that the search/optimization is performed in the policy parameter space?
- More references are needed.
- The parameters of the experiments (number of simulation timesteps, population size, learning rate of PPO/SAC, etc) are not reported.

**Summary Of Recommendation:**

I am willing to change my recommendation to accept if the authors do the followings:
- Either update their claims about using RL for building a locomotion controller, or employ state-of-the-art RL algorithms (e.g., PPO or SAC) to build a robust controller.
- Report the parameters for all training algorithms.
- Add more references, as suggested in the issues.
- Add a detailed discussion about the limitations of the system, showcasing failure cases in the supplemental video.
- Or finally, convince me if I'm wrong about the issues of the paper :)

---

> ### Author Response · Authors · 2021-08-30
> **Response to Reviewer 22ve**
>
> We thank the reviewer for the detailed feedback. We have posted a revision of the paper that includes additional details about experiment setup (Section 4.3 and Appendix C), learning curve (Appendix C) and statistical analysis of all simulation results (Section 5 and Appendix C). The new revision also adds references to the works the reviewer mentioned. Please take a look at the new revision and the diff pdf in supplementary materials.
>
> We provide answers to specific questions below:
>
> > CMA-ES cannot be considered as an RL algorithm.
>
> In this work, we refer to reinforcement learning (RL) as the *problem* we are solving, not as the *method* we are solving it with. That is, we are solving the reinforcement learning *problem* of learning gait transitions, which involves maximizing the return in a MDP. An RL problem can be solved using different techniques, such as policy gradient (e.g. PPO) or gradient-free optimization (e.g. CMA-ES). Our terminology is consistent with definitions in RL textbook (e.g. section 1.4 in [1], chapter 20 in [2]), and RL literature ([3-9]). In the latest revision of the paper, we have made this more clear.
>
> > It is not clear how the reward defined in Eq.(3) always stays positive.
>
> While our reward design does not necessarily guarantee a positive reward at each timestep, the survival bonus does ensure that the reward is positive most of the time, so that the total return of an episode (used in CMA-ES or ARS), or the value function (used in PPO and SAC) is positive. We found that such a reward design is sufficient to prevent the learning algorithm from the local minima of early termination in all our experiments. We have clarified this in the revision.
>
> > The comparison of energy consumption in Fig.(3) is not clear.
>
> Our choice of energy consumption comparison in Fig.(3) is inspired by related works in biology (Fig.2 in [10]), which compares the energy consumption of different gaits at different speeds. It has been observed that animals switch between gaits to minimize energy consumption at each speed, and our result shows that similar results can be obtained using our hierarchical framework. We have modified the related paragraph to clarify our claim.
>
> > The video does not clearly capture fly-trotting gait.
>
> Due to camera position, the air-phase of the fly-trotting gait is not clearly captured in the first part of the video (1:34-1:43). The fly-trotting gait is more clearly visible in the part of the video where the robot responds to abrupt speed changes (2:01-2:14), where the air-phase emerges when the robot is running at 2.5m/s.
>
> > It would be nice to show failure cases.
>
> As reported in the paper, we found our controller to be robust on uneven terrains and abrupt speed profiles. Our controller is less robust on more difficult terrains, such as high stairs and slippery surfaces. Walking robustly on these surfaces will likely require additional information from environment sensors such as cameras and Lidars, and require the high-level policy to modulate foot placement positions in addition to gait timing. As discussed in future works (section 6), our framework is general and can be extended to these scenarios, which we plan to investigate in future studies.
>
> References:
>
> [1] Sutton, Richard and Barto, Andrew. “Reinforcement Learning, An Introduction”
>
> [2] Tedrake, Russ. “Underactuated Robotics”
>
> [3] Ha, David et al. “Recurrent world models facilitate policy evolution”. In NeurIPS 2018
>
> [4] Stanley, Kenneth et al. “Efficient Reinforcement Learning through Evolving Neural Network Topologies”. In GECCO 2002.
>
> [5] Igel, Christian. “Neuroevolution for reinforcement learning using evolution strategies”. In CEC 2003.
>
> [6] Conti, Edoardo et.al. “Improving Exploration in Evolution Strategies for Deep Reinforcement Learning via a Population of Novelty-Seeking Agents”. In NeurIPS 2018
>
> [7] Koutnik, Jan et. al. “Evolving Large-Scale Neural Networks for Vision-Based Reinforcement Learning”. In GECCO 2013
>
> [8] Sehnke, Frank et. al. “Parameter-exploring policy gradients”. In ICANN 2008.
>
> [9] Choromanski, Krzysztof et. al. “Structured Evolution with Compact Architectures for Scalable Policy Optimization”. In ICML 2018.
>
> [10] Hoyt, Taylor et. al. “Gait and the Energetics of Locomotion in Horses.”

---

> > ### Comment · Reviewer_22ve · 2021-09-01
> > **Strong accept**
> >
> > I should thank the authors for their answer and their hard work on updating the paper. I'm really happy with the paper as it is right now, so I've changed my vote to strong accept. It was interesting to see that CMA-ES is outperforming SAC in your experiments.

---

### Official Review · Reviewer_J9Ji · 2021-07-18

**Originality:** Good
**Technical Quality:** Very Good
**Clarity Of Presentation:** Excellent
**Impact:** 4

**Recommendation:**

Strong Accept: I recommend accepting the paper and will argue for my recommendation even if other reviewers hold a different opinion.

**Summary:**

The paper develops energy-efficient controllers for quadrupedal robots.  Energy efficiency is achieved by switching the gait pattern depending on the robot's speed.  To this end, a hierarchical learning framework is introduced.  The gait policy is trained via reinforcement learning, while a body controller optimizes the robot's motion such that the desired velocity can be achieved.  The proposed method is demonstrated in real robot experiments.  The experiments demonstrate the generalization capability of the resultant controller to (i) sharp accelerations and decelerations, (ii) physical perturbations, and (iii) different terrains.  The experiments demonstrate that the resultant controller achieves lower energy across most of the spectrum of examined speeds (0-2.5m/s), when compared to the gait policies "walk", "fly-trot", "slow trot", "rapid trot".

**Issues:**

Please see my comments on where I would make changes, listed in the box "Strengths and Weaknesses".

**Reviewer Expertise:**

Fair: Some knowledge of the area

**Strengths And Weaknesses:**

A.  Strengths:

- In my opinion, the paper is concise and well-organized, being a pleasure to read.  The statement of contributions in the introduction is clear and easy to locate visually.  The related work appears extensive (differentiation of the paper's contributions with related work is also clear).  The paper's figures make it easy to understand the proposed method, and how it compares with existing state-of-the-art methods.

- Proposed method is mostly described in sufficient detail, even for the non-expert reader.

- Real robot experiments appear extensive and are convincing on the generalization capabilities of the proposed method.  Comparison with other state-of-art methods appears extensive.  The accompanying video and supplementary material is similarly supportive.

B.  Where I would make changes:

- I would provide intuition on how the evolutionary strategies for reinforcement learning work.

- I would specify the early termination criteria for infeasible gaits.

- I would quantify statements on the comparative performance of the proposed method (e.g., the statement on line 223).

**Summary Of Recommendation:**

In my opinion, the paper is well-written, with a clear set of technical and experimental contributions, which are supported by (i) a variety of real-world experiments that support the effectiveness of the proposed method (generalization, energy efficiency, and tracking error), and (ii) extensive comparisons with state-of-the-art methods, that further demonstrate the effectiveness of the proposed method.

---

> ### Author Response · Authors · 2021-08-30
> **Response to Reviewer J9Ji**
>
> We thank the reviewer for the thoughtful feedback. We have posted a new revision, in which we addressed the reviewer’s suggestions for improvements. Specifically, we have specified the early termination criteria in section 4.3, added additional experiment details in Appendix C and included learning curves for all experiments. Please take a look at the new revision and the diff pdf in supplementary materials.

---

### Official Review · Reviewer_Zoo3 · 2021-07-22

**Originality:** Very Good
**Technical Quality:** Very Good
**Clarity Of Presentation:** Excellent
**Impact:** 3

**Recommendation:**

Strong Accept: I recommend accepting the paper and will argue for my recommendation even if other reviewers hold a different opinion.

**Summary:**

This paper presents an approach to devising a hierarchical framework for quadruped robots in order to develop an energy efficient controller. The proposed framework combines RL with optimal control that can learn fast and efficient locomotion controllers. The proposed approach consists of a high-level gait generator to be trained using evolutionary strategies formulated as an MDP and a low-level whole body controller using model predictive control with simple dynamics. Numerical simulations demonstrate that the learned controller achieves gait transition depending on the speed of locomotion from walking, trotting to fly-trotting (slow to fast locomotion). In addition, it is demonstrated that the learned policy can be directly transferred to a real quadruped robot without additional data collection or fine-tuning. On the real robot, gait transition and generalization to a variety of new terrains are also demonstrated. These results illustrate the effectiveness of the proposed approach. Furthermore, comparisons to non-hierarchical policies and different learning are presented to evaluate the efficacy of the proposed framework.

**Issues:**

1. There is an impression that the high-level clock-based controller is quite structured. It would be nice if the authors discuss the motivation of this choice. And, also citation of this kind of clock (phased)-based gait generation approach should be included.
2. If I understand correctly, there could be interference between the high-level gait generator and the performance of the low level whole body controller. It would be more informative if this point could be further discussed.
3. The optimized contact forces are converted to joint torques via tau = J_transpose*F as in line 145 on page 4. This relationship generally holds in a quasi-static case. In the dynamic case, inverse dynamics may need to be used. As the locomotion is highly dynamic, I wonder if this quasi-static approximation would be sufficient for successful implementation.
4. In order to demonstrate the effectiveness of the proposed framework, the learning curve should be presented. Also, the number of requires iterations should be provided until convergence.
5. How was the structure of the neural networks determined, i.e., the number of hidden layer and the number of units? Also, how were the weights in the reward function determined? It would be more informative if the information of such a manual tuning process be presented.
6. It is stated that the learned policy in simulations were successfully transferred to the real robot without additional data collection or fine tuning. It would be nice if the authors discuss the reasons why the proposed approach did not require such an additional process in comparison to many of typical RL sim-to-real cases.
7. It is interesting to compare the performance and efficiency against the built-in locomotion controller on the Unitree A1 robot if possible.


**Reviewer Expertise:**

Very good: Comprehensive knowledge of the area

**Strengths And Weaknesses:**

Strength:
This paper is concisely written and easy to read. The proposed framework is able to learn robust and efficient locomotion policy over different speed. The efficiency of the learned controller is evaluated considering the cost of transport. Depending on the speed of locomotion, gait transition can emerge, in which the robot switches from walking, trotting to fly-trotting gaits from slow speed to fast speed. The learned policy in numerical simulations can be directly transferred to a real robot without additional data collection or fine-tuning. Similar gait switching can be achieved on the real robot as well. In addition, comparison to non-hierarchical policies and different learning algorithms suggest the advantages of the proposed approach.

Weakness:
The resultant performance shown in the accompanied video looks quite impressive in terms of robustness of the gait on the robot hardware. To my understanding, the learned controller is confined to straight-line walking, it is not clear how this approach can learn to achieve complex maneuvers such as turning and steering, and possibly stair climbing behaviors which requires some sort of foot placement planning. It would be nice if this paper discusses how the proposed approach can be extended to design controllers for versatile behaviors. It is not clear how much time would be required to learn the behaviors described in this paper. Also, it convergence of the learning algorithm is not very clear. A learning curve during training should be presented to make the paper more convincing and informative to the potential readers.


**Summary Of Recommendation:**

This paper presents a hierarchical approach for optimizing quadruped locomotion by combining high-level RL and low-level whole body controller. The proposed framework is well demonstrated both on numerical simulations and hardware experiments. The paper could be improved by addressing the reviewer's comments.

---

> ### Author Response · Authors · 2021-08-30
> **Response to Reviewer Zoo3 (1/2)**
>
> We thank the reviewer for the constructive feedback. We have posted a new revision of the paper, which includes additional experiment details, such as network architectures (section 4.3 and Appendix C.1), learning curves (Appendix C.1 and C.2), and wall-clock training time (Appendix C.1). Please take a look at the new revision and the diff pdf in supplementary materials.
>
> We provide answers to specific questions below:
> > It is not clear how this approach can learn to achieve complex maneuvers such as turning, steering and possibly stair-climbing.
>
> While we only demonstrated forward walking in the paper, our framework can achieve turning and side walking with a few modifications. As noted in previous works [1], the low-level whole-body controller is already capable of turning and sideway walking. Therefore, to achieve such behaviors, we only need to re-train the high-level gait policy with an extended observation space, which includes desired turning rate or sideway velocity.
>
> For more complex maneuvers such as stair climbing and walking on uneven terrains, a more advanced, terrain-aware foot-placement strategy is necessary, which is beyond the scope of this paper. One way to incorporate this into our framework is to expand the high-level gait policy, so that the state space takes in additional input from environment sensors such as camera and Lidar, and the action space also includes the desired foot placement position. We plan to investigate this approach in future studies.
>
> > The wall-clock time for policy training is not discussed.
>
> One advantage of CMA-ES is that it is highly parallelizable and that the gradient-free policy update is efficient to compute, which reduces the wall-clock training time compared to SAC and PPO. In our experiments, CMA-ES and ARS takes less than 2 hours to reach 1.5 million steps. On the other hand, PPO with multi-processing rollouts takes 10 hours and SAC takes 40 hours. We have included this information in section C.1.
>
> > The motivation and related works of the high-level phase-based gait policy is not clearly discussed.
>
> The choice of phase-based gait policy is mostly inspired by the cyclic nature of locomotion, and has been widely applied in previous works, including classic model-based control [2], learning locomotion [3, 4] and graphics [5]. We have added these works in the revision.
>
> > The interference of the high-level gait policy and low-level whole body controller could be further discussed.
>
> The output of the high-level gait policy could directly affect the stability and energy consumption of the low-level whole-body controller. For example, if the high-level policy outputs an infeasible gait, such as standing with one leg for an extended amount of time, the whole-body controller will fail to complete such a task, the robot will fall, and the episode will be terminated early, and total reward will be reduced. Similarly, if the high-level controller outputs an energy-inefficient gait, such as one involving unnecessary vertical jumps, the whole-body controller will spend unnecessary energy to keep the robot at the desired speed. These interferences are reflected in the reward function, which is used by CMA-ES to optimize the gait policy. See Section 4.2 for more details.
>
> > The conversion from contact forces to motor torques is based on quasi-static assumptions and may not apply in highly dynamic environments.
>
> The use of the Jacobian transpose is indeed based on the quasi-static assumption. We chose to use it because of its simplicity. More importantly, we found it sufficient for all our experiments, including the dynamic fly-trotting gait at 2.5m/s.
>
> > It would be nice to discuss the reason why the proposed approach does not require additional sim-to-real fine-tuning.
>
> One advantage of our hierarchical approach, which combines learning and control, is that learning only happens at the high level, while a lot of the heavy-lifting, including the swing leg control and ground force optimization, is handled by the low-level whole-body controller. Since the whole-body controller can work robustly in the real world, the sim-learned gait policy can be transferred to the real world without additional fine-tuning. See section 5.3 for details.
>
> > It would be interesting to compare the performance and efficiency of the learned controller with A1’s built-in controller from Unitree.
>
> We have contacted the robot manufacturer regarding such possibilities. Unfortunately, the current built-in controller of A1 does not expose low-level information such as joint velocity and joint torques, which makes it difficult to estimate the energy consumption. We will revisit this when we hear more updates from Unitree.

---

> > ### Author Response · Authors · 2021-08-30
> > **Response to Reviewer Zoo3 (2/2)**
> >
> > Reference:
> >
> > [1] Di Carlo et. al. “Dynamic Locomotion in the MIT Cheetah 3 Through Convex Model-Predictive Control.” In IROS 2018.
> >
> > [2] Boussema, Chiheb et. al. “Online gait transitions and disturbance recovery for legged robots via the feasible impulse set”. In RAL 2019.
> >
> > [3] Iscen, Atil et. al. “Policies Modulating Trajectory Generators”. In CoRL 2018.
> >
> > [4] Lee, Joonho et. al. “Learning quadrupedal locomotion over challenging terrain”. In Science Robotics 2020.
> >
> > [5] Holden, Daniel et. al. “Phase-Functioned Neural Networks for Character Control”. In ToG, 2017.

---

> ### Comment · Reviewer_Zoo3 · 2021-09-01
> **Final recommendation**
>
> I would like to thank the authors for their response. The authors have addressed the issues, and revision has been made in a satisfactory manner. I have changed my recommendation to strong accept.

---

### Official Review · Reviewer_ED3s · 2021-07-27

**Originality:** Good
**Technical Quality:** Good
**Clarity Of Presentation:** Good
**Impact:** 4

**Recommendation:**

Weak Accept: I recommend accepting the paper, but will not argue for my recommendation if the majority of other reviewers have a different opinion.

**Summary:**

This paper describes a mix of Model Predictive Control (MPC) and Direct Policy search for quadruped locomotion. The main objective is to allow robots to switch gaits (e.g., walk / trot / gallop) so that the gait is energy-efficient for the target speed. The main idea is that a “whole-body” state-of-the-art MPC (not learned, from previous work) computes (at 500Hz) the joint torques given target contacts and centroidal dynamics, whereas a neural network (optimized in simulation with CMA-ES) computes (at 20Hz) the contacts given the target speed and the current speed. The results show that the resulting “hierarchical controller” works well and does not need to be adapted to transfer to a real robot. The authors claim that end-to-end learning with CMA-ES does not give competitive results. The authors also attempted to learn the high-level policy using PPO, SAC and ARS and they concluded optimizing with CMA-ES gives the best result.


**Issues:**

## Main issue:
- lack of statistical analysis

## Other issues
- Please change CMAES to CMA-ES (the official name: https://en.wikipedia.org/wiki/CMA-ES)
- The MPC is called a “whole-body” controller but it handles legs separately depending on when they are in contact, or not. To me “whole-body” means that the “solution” is computed for all the joint in a single optimization/computation. I would suggest to call this “leg controller” or something similar.
- There are not enough details about the baselines. For instance, for end-to-end, we could suspect that the neural network would be too small, or too big to be optimized by CMA-ES (vs PPO/TRPO/SAC/etc.). But we have no information about the size. Similarly, how was PPO configured? what are the hyper-parameters?
- Releasing the source code would be very helpful, especially for the MPC part

### Minor remark(s):
- The general approach reminded me: https://arxiv.org/abs/2011.04741(Learning Task Space Actions for Bipedal Locomotion) in which the authors combine learning a policy with an inverse dynamics controller (not MPC).

**Reviewer Expertise:**

Good: General knowledge of the area

**Strengths And Weaknesses:**

## Strengths:
- the learning/optimization part compensates the weaknesses of MPC, which makes a very good combination of “classic control theory” and “learning-based robotics”
- convincing results on the videos
- experiments on a real quadruped robot
- comparisons with end-to-end learning and with different learning algorithms

## Weaknesses:
- no statistical analysis of the simulation results (Fig. 3): CMA-ES is a stochastic algorithm: the authors need to run the experiments several times and report median/confidence intervals (otherwise, how do we know that the authors were not lucky once or cherry-picked a particular “seed”?)
- no statistical analysis of the baselines (Table 2, section 5.4): all these algorithms are stochastic too. Also, it would have been good to show “learning curves” in supplementary material, at least to guess when the algorithms have converged (maybe PPO or SAC were stopped prematurely, for instance).
- the authors learned on a simple schedule of target speeds, then say (I am paraphrasing) that it works well for any change of speed and even novel terrains; this needs to be statistically analyses using a “test set” (like in supervised learning) and statistics.


**Summary Of Recommendation:**

I wanted to accept this paper because I agree with the authors that it is very interesting to use learning to improve the state-of-the-art (here MPC) instead of replacing it. This paper takes a good and effective approach and improve it, and I like it.

However, the lack of any statistical analysis as well as the lack of details about the experiments cannot be ignored.

---

> ### Author Response · Authors · 2021-08-30
> **Response to Reviewer ED3s**
>
> We thank the reviewer for the detailed feedback. **We have posted a revision which includes additional details about experiment setup, learning curves and statistical analysis of the results.** Specifically, we added hyperparameters used for CMA-ES, PPO and SAC in Appendix C.1. We also re-run our experiments using 5 random seeds, and updated Fig.3 and Table.2 with statistical analysis. We also included learning curves for the hierarchical policies (Appendix C.1) and non-hierarchical policies (Appendix C.2). Please take a look at the new revision and the diff file in supplementary materials.
>
> We provide answers to other specific questions below:
> > Results about the controller’s generalization to different terrains and different speed profiles should be analyzed statistically.
>
> We have added success rates for the result of generalization to different terrains and different speed profiles. We found that our controller generalizes to scenarios mentioned in the paper at least 80% of the time.
>
> > The naming of “whole-body controller” is misleading.
>
> Thanks for pointing out. Since our low-level controller is based on [1], we have renamed our low-level controller “Convex MPC Controller”, the same term used in [1], to avoid ambiguities.
>
> > A related work in bipedal locomotion (https://arxiv.org/abs/2011.04741) is not discussed.
>
> Thanks for pointing this out. We have included a discussion of this paper in Related Works.
>
> > Releasing the source code.
>
> We will open-source the code, including the high-level gait policy, low-level convex MPC controller and the robot interface once the paper is published.
>
> References:
>
> [1] Di Carlo et. al. “Dynamic Locomotion in the MIT Cheetah 3 Through Convex Model-Predictive Control.”  In IROS 2018.

---

> > ### Comment · Reviewer_ED3s · 2021-09-02
> > **New version**
> >
> > Thank you for the updated version. I moved my recommendation to weak accept now that there are statistics.
> >
> > For the next version/next paper, please have a look at:
> > - https://arxiv.org/abs/1806.08295
> > - https://arxiv.org/abs/2108.13264
> >
> > To improve your statistical comparisons. In addition, 5 replicates is very low to get statistics, and you should aim for a higher number in future work.

---

### Author Response · Authors · 2021-09-01
**[To All] Brief Summary of Response**

This is a copy of our response to the meta review from AC, reproduced here for visibility.

--------------------------------------------Original Content--------------------------------

We thank the reviewers for their thoughtful comments, and the AC for the review summary. **We have uploaded a revision of the paper. We have also included a pdf in supplementary materials, which contains the diff of the revision compared to the original submission.** Please take a look. We summarize the changes below:

1. **Added experiment details and statistical analysis.**

We have added additional details about the experiment setup for CMA-ES, PPO, SAC (in Section 4.3 and Appendix C.1), and added details about non-hierarchical environment setups in Appendix C.2. We also carefully tuned the baselines for their optimal performance, and conducted additional experiments to verify the statistical significance of our results.

In summary, we found that our method still significantly outperformed most of the baselines. Specifically, the hierarchical policy trained by CMA-ES is consistently more energy efficient than hand-tuned gaits, and learns a more stable gait than non-hierarchical policies. Moreover, CMA-ES still significantly outperforms PPO and SAC in our setup. Please find the corresponding learning curves in Fig.11 and Fig.12. Fig.3 and Table.2 are also updated with statistics.

2. **Addressed the ambiguity between CMA-ES and reinforcement learning**

As stated in our response to reviewer 22ve, in this work, we refer to reinforcement learning (RL) as the *problem* we are solving, not as the *method* we are solving it with. That is, we are solving the reinforcement learning *problem* of learning gait transitions, which involves maximizing the return in a MDP. An RL problem can be solved using different techniques, such as policy gradient (e.g. PPO) or gradient-free optimization (e.g. CMA-ES). Our terminology is consistent with definitions in RL textbook (e.g. section 1.4 in [1], chapter 20 in [2]), and RL literature ([3-9]). In the latest revision of the paper, we have made this more clear.


3. **Renamed “whole-body controller” as “Convex MPC Controller” to avoid ambiguity.**

As pointed out by reviewer ED3s, our low-level controller is based on a simplified dynamics model and does not consider full robot dynamics. To avoid confusion, we have renamed our low-level controller “Convex MPC Controller”, to be consistent with the original paper [10].

4. **Added references**

Following suggestions from reviewer ED3s and 22ve, we have added the following papers into the related works section:
* Learning Task Space Actions for Bipedal Locomotion
* DeepMimic: Example-Guided Deep Reinforcement Learning of Physics-Based Character Skills
* Phase-Functioned Neural Networks for Character Control
* SIMBICON: Simple Biped Locomotion Control


References:

[1] Sutton, Richard and Barto, Andrew. “Reinforcement Learning, An Introduction”

[2] Tedrake, Russ. “Underactuated Robotics”

[3] Ha, David et al. “Recurrent world models facilitate policy evolution”. In NeurIPS 2018

[4] Stanley, Kenneth et al. “Efficient Reinforcement Learning through Evolving Neural Network Topologies”. In GECCO 2002.

[5] Igel, Christian. “Neuroevolution for reinforcement learning using evolution strategies”. In CEC 2003.

[6] Conti, Edoardo et.al. “Improving Exploration in Evolution Strategies for Deep Reinforcement Learning via a Population of Novelty-Seeking Agents”. In NeurIPS 2018

[7] Koutnik, Jan et. al. “Evolving Large-Scale Neural Networks for Vision-Based Reinforcement Learning”. In GECCO 2013

[8] Sehnke, Frank et. al. “Parameter-exploring policy gradients”. In ICANN 2008.

[9] Choromanski, Krzysztof et. al. “Structured Evolution with Compact Architectures for Scalable Policy Optimization”. In ICML 2018.

[10] Di Carlo et. al. “Dynamic Locomotion in the MIT Cheetah 3 Through Convex Model-Predictive Control.” In IROS 2018.

---

### Meta-Review · Area_Chair_Zmix · 2021-08-04

**Recommendation:** Accept (Oral)
**Confidence:** 3

**Metareview:**

There was a wide range of reviewer opinions on this paper. Most reviewers were in agreement that the restricted policy space used by the authors was chosen well and led to compelling results in the experimental section and the supplementary video. The main criticisms of the paper were a lack of detail about the methods employed in experiments as pointed out by several reviewers. There is no data on how the CMA-ES algorithm was actually run, and no learning curves or statistics of repeated testing were reported for the baseline methods. If the paper were to be accepted, the authors must do a substantial job revising the experimental description to provide these details. Moreover, as pointed out by reviewer 22ve the claim that their technique is "RL" should be rewritten since CMA-ES is a traditional metaheuristic optimization technique. The authors can claim they are "learning" using simulated experience, but "RL" is certainly a misnomer.

After receiving the author feedback, the reviewers are satisfied with the changes made to the paper, and are now in agreement that the paper should be accepted.

---

> ### Author Response · Authors · 2021-08-30
> **[To All] Brief Summary of Author Response**
>
> We thank the reviewers for their thoughtful comments, and the AC for the review summary. **We have uploaded a revision of the paper. We have also included a pdf in supplementary materials, which contains the diff of the revision compared to the original submission.** Please take a look. We summarize the changes below:
>
> 1. **Added experiment details and statistical analysis.**
>
> We have added additional details about the experiment setup for CMA-ES, PPO, SAC (in Section 4.3 and Appendix C.1), and added details about non-hierarchical environment setups in Appendix C.2. We also carefully tuned the baselines for their optimal performance, and conducted additional experiments to verify the statistical significance of our results.
>
> In summary, we found that our method still significantly outperformed most of the baselines. Specifically, the hierarchical policy trained by CMA-ES is consistently more energy efficient than hand-tuned gaits, and learns a more stable gait than non-hierarchical policies. Moreover, CMA-ES still significantly outperforms PPO and SAC in our setup. Please find the corresponding learning curves in Fig.11 and Fig.12. Fig.3 and Table.2 are also updated with statistics.
>
> 2. **Addressed the ambiguity between CMA-ES and reinforcement learning**
>
> As stated in our response to reviewer 22ve, in this work, we refer to reinforcement learning (RL) as the *problem* we are solving, not as the *method* we are solving it with. That is, we are solving the reinforcement learning *problem* of learning gait transitions, which involves maximizing the return in a MDP. An RL problem can be solved using different techniques, such as policy gradient (e.g. PPO) or gradient-free optimization (e.g. CMA-ES). Our terminology is consistent with definitions in RL textbook (e.g. section 1.4 in [1], chapter 20 in [2]), and RL literature ([3-9]). In the latest revision of the paper, we have made this more clear.
>
>
> 3. **Renamed “whole-body controller” as “Convex MPC Controller” to avoid ambiguity.**
>
> As pointed out by reviewer ED3s, our low-level controller is based on a simplified dynamics model and does not consider full robot dynamics. To avoid confusion, we have renamed our low-level controller “Convex MPC Controller”, to be consistent with the original paper [10].
>
> 4. **Added references**
>
> Following suggestions from reviewer ED3s and 22ve, we have added the following papers into the related works section:
> * Learning Task Space Actions for Bipedal Locomotion
> * DeepMimic: Example-Guided Deep Reinforcement Learning of Physics-Based Character Skills
> * Phase-Functioned Neural Networks for Character Control
> * SIMBICON: Simple Biped Locomotion Control
>
>
> References:
>
> [1] Sutton, Richard and Barto, Andrew. “Reinforcement Learning, An Introduction”
>
> [2] Tedrake, Russ. “Underactuated Robotics”
>
> [3] Ha, David et al. “Recurrent world models facilitate policy evolution”. In NeurIPS 2018
>
> [4] Stanley, Kenneth et al. “Efficient Reinforcement Learning through Evolving Neural Network Topologies”. In GECCO 2002.
>
> [5] Igel, Christian. “Neuroevolution for reinforcement learning using evolution strategies”. In CEC 2003.
>
> [6] Conti, Edoardo et.al. “Improving Exploration in Evolution Strategies for Deep Reinforcement Learning via a Population of Novelty-Seeking Agents”. In NeurIPS 2018
>
> [7] Koutnik, Jan et. al. “Evolving Large-Scale Neural Networks for Vision-Based Reinforcement Learning”. In GECCO 2013
>
> [8] Sehnke, Frank et. al. “Parameter-exploring policy gradients”. In ICANN 2008.
>
> [9] Choromanski, Krzysztof et. al. “Structured Evolution with Compact Architectures for Scalable Policy Optimization”. In ICML 2018.
>
> [10] Di Carlo et. al. “Dynamic Locomotion in the MIT Cheetah 3 Through Convex Model-Predictive Control.” In IROS 2018.

---

### Decision · Program_Chairs · 2021-09-13

**Decision:**

Accept (Oral)

**Comment:**

There was a wide range of reviewer opinions on this paper. Most reviewers were in agreement that the restricted policy space used by the authors was chosen well and led to compelling results in the experimental section and the supplementary video. The main criticisms of the paper were a lack of detail about the methods employed in experiments as pointed out by several reviewers. There is no data on how the CMA-ES algorithm was actually run, and no learning curves or statistics of repeated testing were reported for the baseline methods. If the paper were to be accepted, the authors must do a substantial job revising the experimental description to provide these details. Moreover, as pointed out by reviewer 22ve the claim that their technique is "RL" should be rewritten since CMA-ES is a traditional metaheuristic optimization technique. The authors can claim they are "learning" using simulated experience, but "RL" is certainly a misnomer.

After receiving the author feedback, the reviewers are satisfied with the changes made to the paper, and are now in agreement that the paper should be accepted.